# Length Dependence of Vocabulary Richness

**Niklas Zechner**

Språkbanken

University of Gothenburg

`niklas.zechner@gu.se`

## Abstract

The relation between the length of a text and the number of unique words is investigated using several Swedish language corpora. We consider a number of existing measures of vocabulary richness, show that they are not length-independent, and try to improve on some of them based on statistical evidence. We also look at the spectrum of values over text lengths, and find that genres have characteristic shapes.

## 1 Introduction

Measures of lexical richness have several uses, including author identification, other forms of text classification, and estimating how difficult a text is. One of the simplest and most obvious measures of lexical richness is to compare the size of the vocabulary (that is, how many different words) to the size of the text (how many words in total). This can be done in several ways, most straightforwardly as the type-token ratio (henceforth TTR), $u/n$, where $u$ is the number of unique words (types) and $n$ is the total number of words (tokens). Thus, for the sentence "this example is this example", there are three types and five tokens, so TTR is $u/n = 3/5 = 0.6$.

The obvious problem with TTR is that it changes with the length of the text. As we write a text, the more words we have already written, the more likely it is that the next word will be one that has already been used, so TTR goes down as the text grows longer. Many attempts have been made to transform this measure into something independent of the length of the text, but many of those attempts were made in an age before "big data", or even before computers, and were based on a priori reasoning rather than statistical analysis (Tweedie and Baayen, 1998).

We will start by looking at some of these measures, and test them on a set of corpora to see how they hold up for a wide range of different $n$. After comparing some of the previous methods, we will briefly look into using the empirical data to come up with a better suggestion. The results give rise to another question: What if instead of aiming for a length-independent measure, we consider *how* the values change with the length? Can that actually tell us new and interesting things?

We find that if we analyse the type count for different sample lengths, we see clear and consistent differences between different types of text. This may be useful for genre classification, or for a more detailed description of the text complexity.

Although these measures are usually applied to specific texts, we here apply them to entire corpora. We will discuss the effects of this after seeing the results.

## 2 Data

Språkbanken (the Swedish Language Bank) at the University of Gothenburg (spraakbanken.gu.se) has a large collection of text corpora, mainly in Swedish but including several other languages. In this study, we use Swedish texts, focusing on large and homogeneous corpora, listed in the appendix.

We extract the type count $u$ for several different lengths $n$. Words are case-independent but otherwise counted as written, without lemmatisation. For each $n$, we divide the corpus in chunks of length $n$, dropping any overflow at the end, and take the mean value of $u$ for each of these chunks. (In some cases we remove the last value for being an outlier; presumably this is because it is the only value where a large part of the data is dropped due to overflow.) We use a pseudo-logarithmic scale for ease of reading, extracting values for $n = 10, 20, 50, 100, 200, 500, 1000...$ up to the maximum possible for each corpus; the largest go up to 500 million tokens.

## 3 Testing existing measures

First of all, we can test and verify that TTR does go down. Figure 1 shows TTR for 31 corpora.

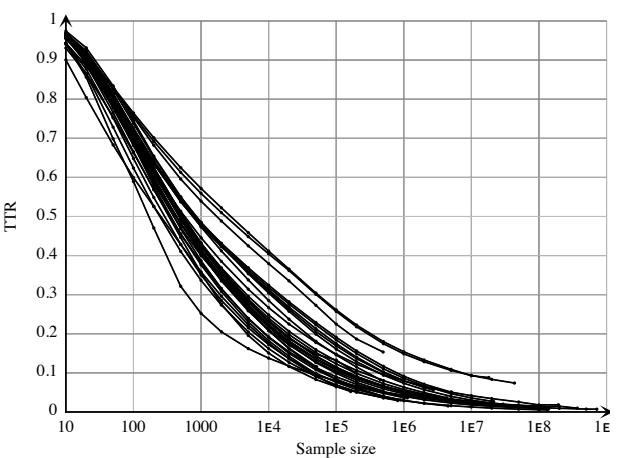

Figure 1: Type-token ratio

It seems likely that, as we compare different-size corpora, effects of size changes might be best described in terms of multiplicative changes rather than additive, so we might try looking at the logarithms of $n$ and $u$. We see in Figure 2 that the result looks fairly close to a straight line.

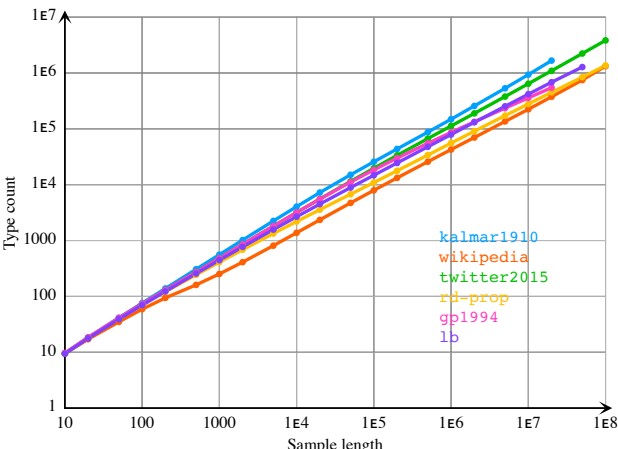

Figure 2: Type count

The first obvious method, then, is to assume that this is indeed a straight line, and use the slope of that line as our presumed length-independent measure of richness, that is, $\log u / \log n$. This was proposed by Herdan (1964). We see in Figure 3 that the measure is decreasing quite steadily for all the texts. The six corpora used here are chosen

partly for being large, and partly for having large differences in type count; many other corpora are not nearly as well separated.

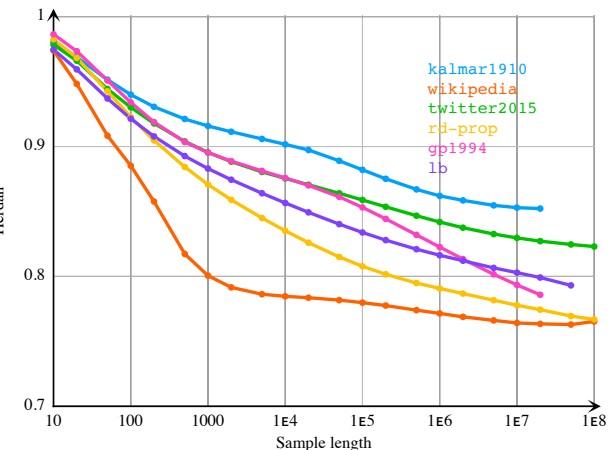

Figure 3: Herdan's measure

Let us pause for a moment and consider what this figure illustrates. The fact that the measure decreases is not in itself a problem; although we are aiming for a near-constant, we should not expect it to be perfect. The amount of variation is also not relevant; we could change that by adding or multiplying by a constant. Regardless of how large the variation is, we would also change the axes of the graph, so a glance at the variation of a single curve in the graph does not tell us whether the measure is near-constant in a relevant sense.

What actually matters is comparing the curves. If the measure is to reliably compare different texts, regardless of the (sample) size for each text, what we need is to have the lines separated insofar as possible. If the lowest point of curve A is higher than the highest point of curve B, then we have successfully determined that A has a higher richness. We should also keep in mind that the first few points of the curve are not as important – we are probably not very interested in measuring richness for very short texts, so although the graphs go all the way from 10, we can mostly ignore values below 1000 or so. We would be content if the measure can separate the lines from that point on.

As we see in Figure 3, this is not quite the case here. This measure works better than TTR, but the curves are still close enough that their ranges overlap. We will compare with a few other measures.

Guiraud (in 1954, as cited by Hultman and Westman (1977)) proposed the measure $u/\sqrt{n}$,

shown in Figure 4. This does not separate the curves particularly well, and does not seem to have any advantage over the previous method.

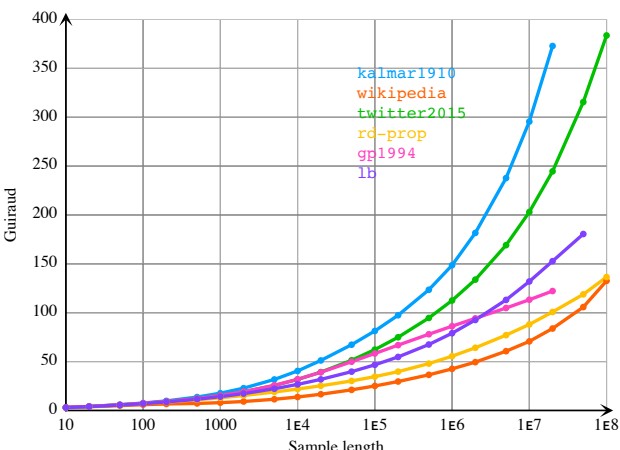

Figure 4: Guiraud's measure

Dugast (1979) built on Herdan by suggesting $\log u / \log \log n$, seen in Figure 5. We find no advantage with this method, and only added conceptual complexity with the double logarithm.

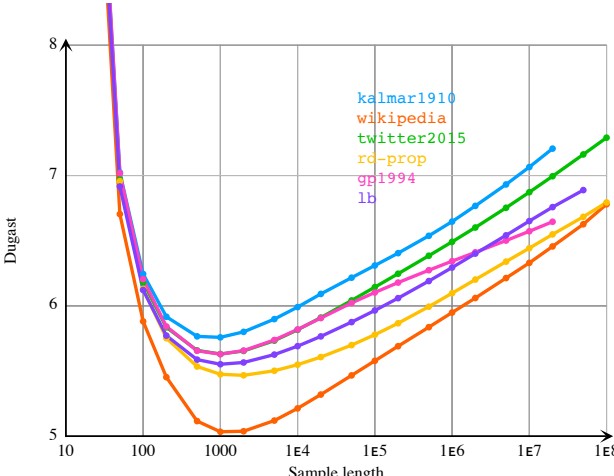

Figure 5: Dugast's measure

Brunet (1978) proposed $n^{\wedge}(u^{-a})$, where usually $a = 0.172$. This is shown in Figure 6. This too is a fairly conceptually complicated method which shows no sign of improving the results.

Maas (1972) found another approach, with $(\log n - \log u)/(\log n)^2$, see Figure 7. This seems marginally more effective at separating the curves.

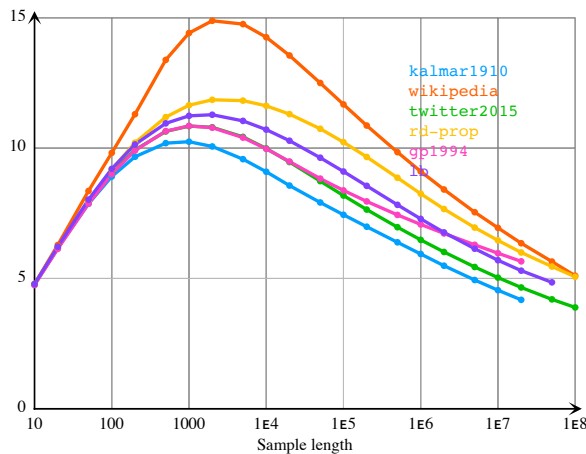

Figure 6: Brunet's measure

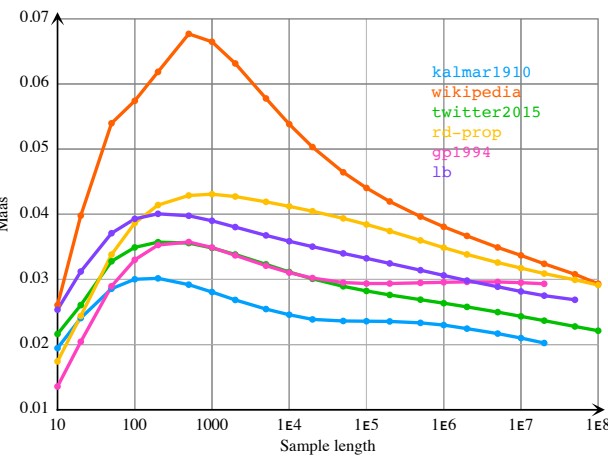

Figure 7: Maas's measure

Hultman and Westman (1977) defined the OVIX measure as

$$\frac{\log n}{\log \left( 2 - \dfrac{\log u}{\log n} \right)}$$

which is seen in Figure 8. This is a measure commonly used in Sweden, including by Språkbanken. As we see, this also does a passable job, but there is a clear rising trend for most curves. This is confirmed by further testing on other corpora.

## 4 Improving measures

By analysing the way these measures depend on $n$, we may be able to adjust and improve them. As noted, the fact that the curve of $\log u$ against $\log n$ is close to a line suggests that $(\log u / (\log n)$ may

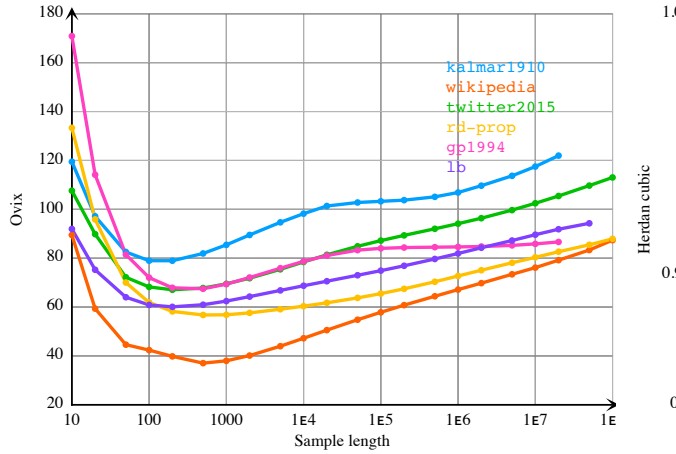

Figure 8: Ovix

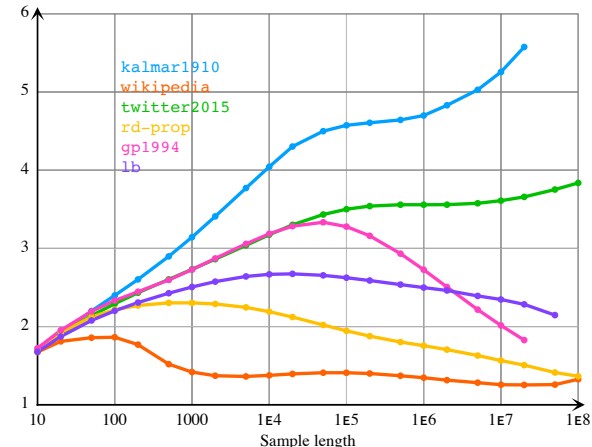

Figure 11: Adjusted Guiraud

be a constant, as per Herdan. But that assumes that the line passes through $(0, 0)$; if the line passes though $(0, m)$ for some $m$, we should expect that $(\log u - m)/\log n$ is constant. We find that for a subset of the corpora, the best-fitting line gives $m = 0.4$, and we see in Figure 9 that $(\log u - 0.4)/\log n$ does look a lot flatter. As before, we pay less attention to the values where $n < 1000$.

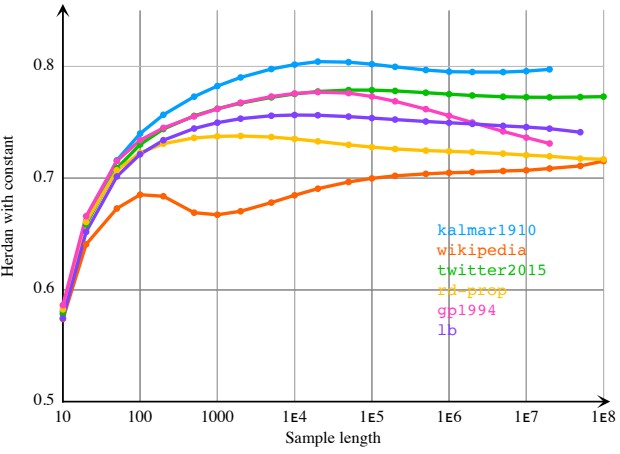

Figure 9: Herdan with constant term

On the other hand, we know that a text with one word certainly also has one unique word, so logically the curve of $\log u$ against $\log n$ *must* pass though $(0, 0)$. Empiricism is all good and well, but if we want results that hold up for other data, perhaps we are better off not violating basic logic. What if instead of a line, we fit the points to a polynomial curve with zero constant term? Trying

second, third and fourth order polynomials suggests that third is a good compromise. We find the best fit for six corpora, take the average for the quadratic and cubic terms, and get the adjusted measure

$$\log u/\log n + 0.044(\log n) - 0.0024(\log n)^2$$

You can see in Figure 10 that this separates the curves considerably better than the pure Herdan measure. From looking at the graph, this is probably the best option we have here, but we should note that the coefficients vary quite a bit between corpora (standard deviations are 0.015 and 0.0017), so this is not universal enough to adopt as some sort of standard measure.

We can also consider the Guiraud approach, and try to adjust it. We notice that while TTR $(u/n)$

goes steadily down, Guiraud ($u/n^{0.5}$) goes up. Perhaps we can find a middle ground? Figure 11 shows the results for $u/n^{0.75}$, which looks overall much flatter and better separating the curves. This may not be a better result than the previous one, but it does have the advantage of not depending on experimentally determined coefficients.

## 5 Fixed-length measures

Is there another option, using only the length and the type count? Yes, there is an option which is in principle completely independent of text length: Measure the type count (or equivalently TTR) for a fixed length. One option would be to measure only the first $n$ words of a text, but that could mean that a small part of the text has a large impact, so probably a better method is to cut the text into pieces of length $n$ and take the average, exactly as we have done above.

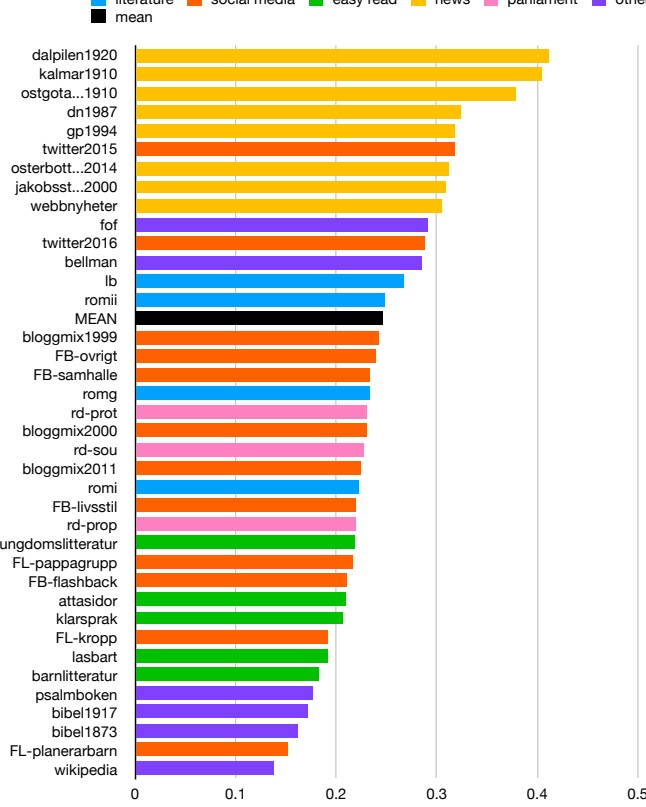

Figure 12: TTR at $n = 10\,000$

Figure 12 shows the results for $n = 10\,000$, on 38 corpora. We see that it fairly well separates several categories of text. The eight newspaper corpora are above all but one other, with the three oldest getting the highest value, followed by the two from the late 1900s, then the two from printed

newspapers in 2000 and 2014, and last the web-based news texts. (The difference may be partially explained by OCR errors.) The social media and blog texts are a little more scattered, but all below the mean, except Twitter, which in both cases is higher. The four corpora of novels are not quite the same level, but all higher than all of the ones in the "easy read" category. In that category, young adult literature is the highest and children's literature the lowest. Parliamentary data is all below the mean but above "easy read". Near the bottom we find, perhaps surprisingly, the Bible, along with Wikipedia, neither of which are primarily known to be easy reads. Altogether, these results should tell us that this is at least a meaningful measure.

That leaves the question of choosing an $n$. Very low values might give strange effects, very high values would make it unusable for shorter texts. Other values were tested for comparison: $n = 10$ gives little useful information, while $n = 100$ ranks all the novels below most of social media, and beyond that we get mostly unremarkable results from just looking at the ranking. Based on these limited results, $n = 10\,000$ is a good choice, and for short texts we can settle for $n = 1000$.

## 6 Spectrum comparison

Instead of considering type counts for only one $n$, what if we measure for many values of $n$, and look at the whole spectrum? This is essentially what we already did in all of section 3, and we could see that the curves for the different corpora certainly did have different shapes – some of them even crossed each other, which implies that any one number is not going to tell us the whole truth.

To compare corpora instead of methods, we need to pick one method, one way to transform $u$ based on $n$. Using plain TTR as seen in Figure 1 would make it difficult to tell the difference between shapes, and picking one of the tested methods seems like too arbitrary a choice. So for the purposes of this section, we will evade the problem. We normalise the type count (or equivalently TTR) for each $n$ by subtracting the mean and dividing by the standard deviation. That is, the values on the vertical axis are in terms of standard deviations above the mean, counted for each separate value on the horizontal axis. (For high values, the mean/sd change erratically because of corpora dropping off. We adjust the normalisation to gradually change from actual to extrapolated mean/sd.)

Figures 13-22 show the spectra for each category. Some curves are shorter because of limited data. Figures 13-15 show different types of web-based texts, one set of blog texts and two different internet forums. We can see that each category is a little different, but all the curves share some characteristics – a short rise, then a drop, then flatter, and finally a small rise. Most of them start slightly above the mean, and end below the mean.

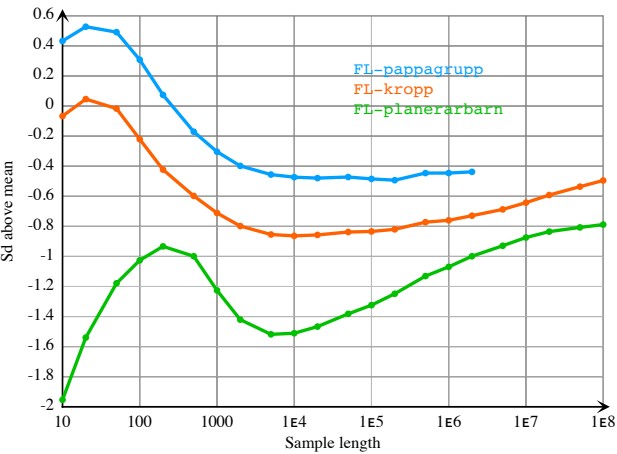

Figure 15: Spectrum for the *Flashback* forum

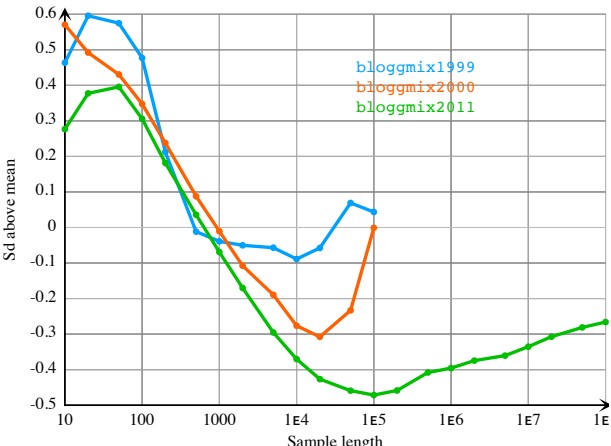

Figure 13: Spectrum for blog texts

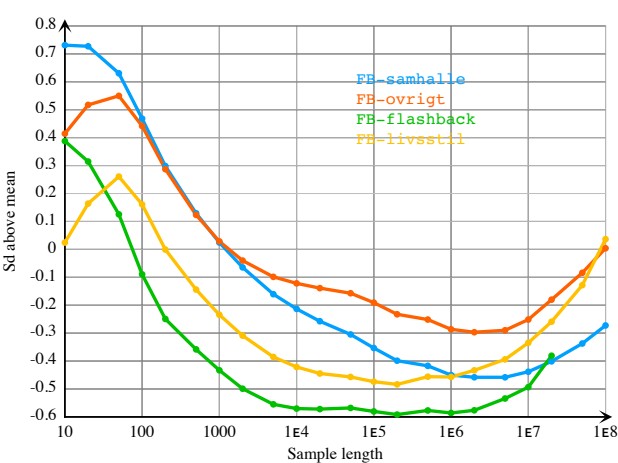

Figure 14: Spectrum for the *Familjeliv* forum

Figure 16 shows the "easy read" category. Despite being unrelated, the curves share the same shape, which is clearly different from the web-based corpora – a drop, then a rise, peaking around 1000 without reaching the mean, then a drop.

Figures 17-18 show news texts, with Figure 17 showing three newspapers from the early 1900s,

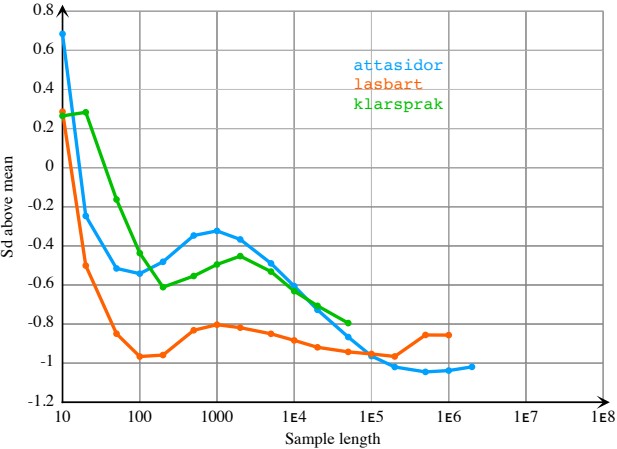

Figure 16: Spectrum for easy-read texts

and Figure 18 showing four more recent newspapers and one web-based news corpus. As with the blog/forum collection, we see that these two related categories have clear similarities: a slow rise up to between 10 000 and 100 000, and then a drop. But they are also visibly distinct, with the older newspapers having higher values and rising near the end. Aside from some unpredictable behaviour for $n < 1000$, the curves in each category are remarkably similar in both shape and level.

Figures 19-20 show literary texts, with Figure 19 showing regular novels and Figure 20 showing children's fiction and young adult fiction. They are all comparatively straight and dropping slightly. Children's literature is generally lower than young adult literature, and they both drop faster than the curves for books aimed at adults.

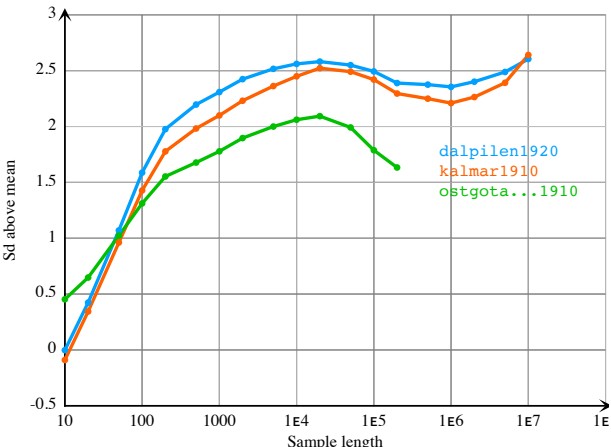

Figure 17: Spectrum for old newspapers

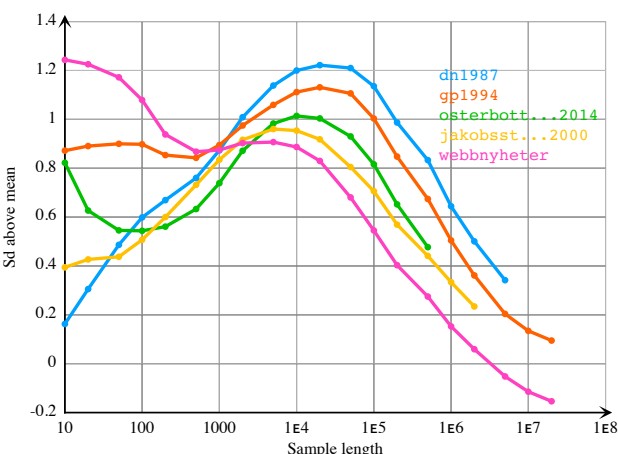

Figure 18: Spectrum for recent newspapers

complex sentences with few function words; for medium values (around 10 000), complexity in topics, with many names etc.; and for high values, variety in topics. This may explain why newspapers peak in the middle (they address complex topics with many names, but return to the same topics), social media drop in the middle (they address simpler topics but with a wider variety in topic and style), and youth novels go down (they are on simple topics and consistent for entire books). Further speculation is left for future work.

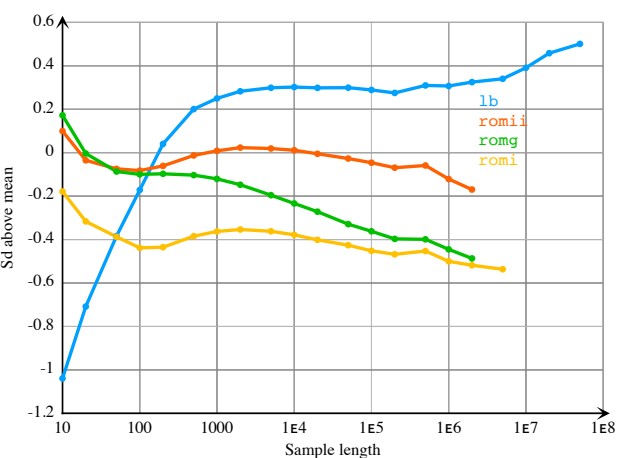

Figure 19: Spectrum for novels

Figure 21 shows religious texts. We see two translations of the Bible, with very similar curves – both dropping, rising, levelling out, but unlike the easy read category they level out at about the same level where they started. Also included is a book of church hymns, which happens to level out at a similar level, but starts with a large rise.

Finally, in Figure 22, we see three uncategorised corpora – one from a 1700s songwriter, one from a popular science magazine, and one from Wikipedia. As expected, they show very different shapes and levels, and are clearly distinct from each other as well as all the other curves.

Explanations of the shapes are tentative at this point, but we can guess at the meaning of high richness in different regions of sample length: For low values (roughly 100-1000), it may indicate

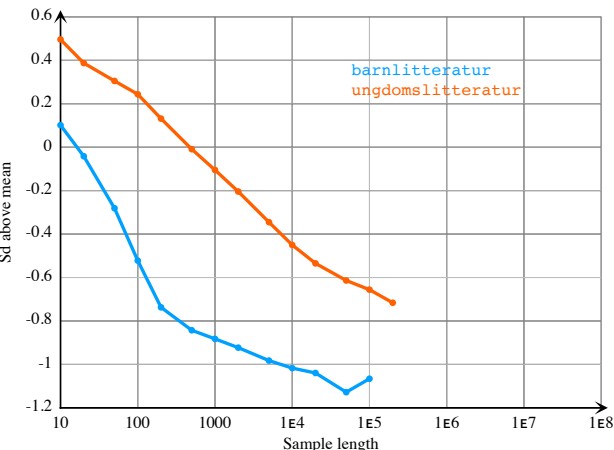

Figure 20: Spectrum for youth novels

## 7 Applicability

Is it reasonable to apply measures like these on an entire corpus instead of just separate texts? First, "separate texts" is not necessarily well defined. Is

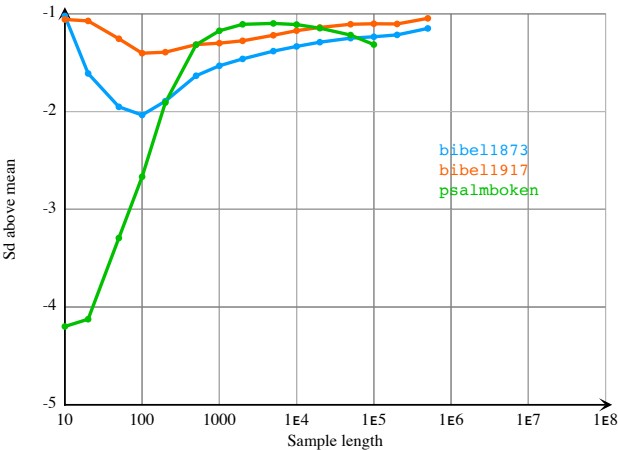

Figure 21: Spectrum for religious texts

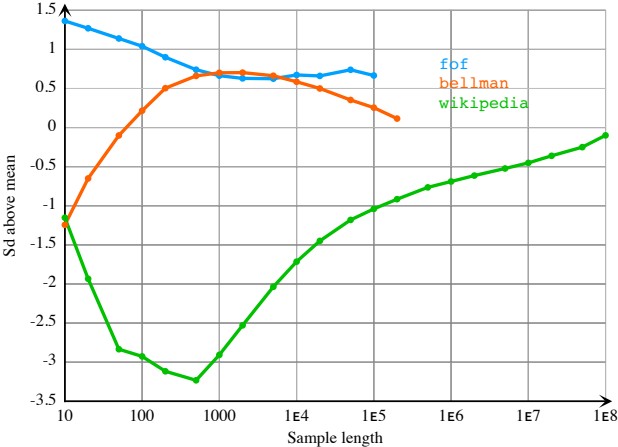

Figure 22: Spectrum for some other texts

a newspaper one text, or each article? Books in a series? Multiple entries on a web page? Second, for low values of $n$, running the entire corpus at once should make little difference. For example, if $n = 100$ and the typical length of a text is $10\,000$, only about 1% of samples would contain two texts, and the rest only one. For high values of $n$, using only separate texts would leave us with no data at all – it would be difficult to find singular coherent texts spanning hundreds of millions of words. This means that allowing corpora of multiple authors and topics is our only option for large $n$.

But we can also look at the results. Are the differences between the curves largely caused by differences in text length? If that was the case, we would expect that when a curve reaches the "critical $n$" where we go from a single text to multi-

ple texts, the vocabulary richness should increase rapidly. The curve we would expect to see is one that starts out mostly flat (because hardly any texts are that short), then slowly decreases (as others reach their critical $n$ and bring up the mean), then rapidly jumps up as it reaches its critical $n$, and then slowly decreases again. This is not a pattern that we see anywhere, so we can conclude that text length is not the driving factor of the curve shapes.

# 8 Conclusion

The task of finding a length-independent measure of vocabulary richness is difficult at best. We have seen that many traditional measures are not satisfactory, and made some suggestions as to how they can be improved. Perhaps the most obvious approach is to use average TTR over a sample length, with $10\,000$ words being a good sample length.

The figures show that the curves have very different shapes, and often cross. Thus, the ranking of corpora changes depending on the length of the text sample, so a perfect solution is not possible, or at least cannot be expressed as a single number.

Is this spectrum method useful for genre classification? It is perhaps rare that we need to analyse entire hundred-million-word corpora to see if they are made up of novels or newspapers, but we do see some differences even for much smaller lengths. We have also gained insight into what makes it difficult to find a good measure of vocabulary richness. Most importantly, we have seen that there are notable differences between genres, and raised for future research the question of why.

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

## Appendix A. List of corpora

The following corpora were used, all of which can be found at spraakbanken.gu.se/en/resources (some only in scrambled versions). All texts are in Swedish.

**attasidor** issues of the newspaper *8 sidor* in easy Swedish

**barnlitteratur** collection of children's literature

**bellman** lyrics from the Swedish songwriter C. M. Bellman (1740-1795)

**bibel1873** full text of the 1873 Swedish Bible translation

**bibel1917** full text of the 1917 Swedish Bible translation

**bloggmix1999** collection of blog texts from 1999

**bloggmix2000** collection of blog texts from 2000

**bloggmix2011** collection of blog texts from 2011

**dalpilen1920** issues of the newspaper *Dalpilen* from the 1920s

**dn1987** issues of the newspaper *Dagens Nyheter* from 1987

**familjeliv (FL) kropp** webforum *familjeliv.se*, subforum about the human body

**familjeliv (FL) planerarbarn** webforum *familjeliv.se*, subforum about planning to have children

**familjeliv (FL) pappagrupp** webforum *familjeliv.se*, subforum for fathers

**flashback (FB) flashback** webforum *flashback.se*, subforum about the forum itself

**flashback (FB) livsstil** webforum *flashback.se*, subforum about lifestyle

**flashback (FB) ovrigt** webforum *flashback.se*, subforum about miscellaneous topics

**flashback (FB) samhalle** webforum *flashback.se*, subforum about society

**fof** issues of the popular science magazine *Forskning & Framsteg*

**gp1994** issues of the newspaper *Göteborgsposten* from 1994

**jakobstadstidning2000** issues of the newspaper *Jakobstads Tidning* from 2000

**kalmar1910** issues of the newspaper *Kalmar* from the 1910s

**klarsprak** administrative authority texts

**lasbart** collection of easy-read texts and children's books

**lb** the Swedish Literature Bank, a collection of literature mainly from around 1900

**osterbottenstidning2014** issues of the newspaper *Österbottens Tidning* from 2014

**ostgotaposten1910** issues of the newspaper *Östgötaposten* from the 1910s

**psalmboken** the hymn book of the Church of Sweden

**rd-prop** Swedish parliament texts, propositions

**rd-prot** Swedish parliament texts, protocols

**rd-sou** Swedish Government Official Reports

**romg** collection of older novels

**romi** collection of modern novels

**romii** collection of modern novels

**twitter-2015** posts from twitter.com, 2015

**twitter-2016** posts from twitter.com, 2016

**ungdomslitteratur** young adult literature

**webbnyheter2005** collection of online newspaper texts from 2005

**wikipedia** Swedish Wikipedia, collected in 2017