# OpenReview forum: "Length Dependence of Vocabulary Richness"
_NoDaLiDa/2023/Conference — NoDaLiDa 2023_

### Official Review · Reviewer_ETYk · 2023-02-16
**The article is interesting, clearly deviates [sic] from your standard lg tech paper anno this spring and should be accepted**

**Rating:** 7
**Confidence:** 3

**Review:**

The term "word" is not defined. The authors should at the outset specify whether they mean "wordform" or "lexeme", and in the former case, whether they count "björn", "Björn" (sentence-initially) and "Björn" (the name, i.e., not sentenc-initially) as one, two or three types. Contrary to many other language technologists, the ones at Språkbanken are able to make these distinctions, we thus want to know what you did.

The reference to "Språkbanken" (line 065) is by the name only. Inititated readers may combine this word with the word "Swedish" on line 015 and end up in Gothenburg. For other readers this exercise may prove too difficult. The proper reference omces at line 089, move the parenthesis to the first mention.

As the discussion shows, the study of type/token frequencies has a long history. It has also been neglected in recent years, which means that earlier authors have not had access to nearly as large and well-curated corpora as the present study has. The basic measure is simple and the problem seemingly trivial. Still, the article has been able to show hitherto unseen genre differences and thus been able to raise a new question of research. This rare achievement makes it deserving a place at NoDaLiDa.

The page limit of 8 is brutal and short, the moments raised in the following may be disregarded for reasons of space (at least to the extent that they require more than 5 additional lines). I still want to mention them, as they will improve the article (or its announced follow-up).

The corpora are named, but the names are not explained. The initiated reader recognises Göteborgsposten (gp) Dagens Nyheter (dn), guesses on the basis of colour that kalmar is some newspaper, and recognises some other newspapers as well. Labels "fof, lb, romi, romg, attasidor" are incomporehensible (for fof we get a cue on line 698). rd is probably Riksdagen. In short, we need an explanation.

Line 442 to 511 should be singled out as a separate chapter, as it stands out as different from the former ones..

Except for the short discussion on lines 798-809, not much is said on what the different shapes actually express. Several of the early approaches (e.g. Dugast, Brunet) are dismissed as conceptually complicated without no sign of improving the results. Now, the first of these two points may also be said about the authors' approach on pp 6ff. This may of course be defended if they do improve the results. I still would have liked to see the rationale behind normalising the type count and counting them for each separate value on the orizontal axis (lines 530ff). Yes, we see that this groups curves in 8 different shape classes, whereas the same can probably not be said for the approaches on page 2-4 (I say "probably", since the corpora are only partly the same in these two sections). Still, the earlier curves are really uniform, contrary to the genre groups in Figures 13 - 20. Fusing them into one graph would not have given anything resembling figures 1-11.

What we see (7-8 genre classes, each with its own shape) is quite clear. The reason why this is so is suggested for future research. My suggest for that endeavor would be to start out with asking what high and low y axis values for the different values of x really tell us. Had space made it possible, the paper should have at least thrown in some thoughts on this.

Modern language technologists do evaluations rather than analyses, a possible one would thus be to throw in some blind genres and look at whether they are reliably classified (with the usual recall/precision battery). One result in point may be "lb" in Figure 19: Based upon shape we would classify it as an old newspaper and not as a novel. Another is Psalmboken, straddling the class of old newspapers. But for the other shapes (= the vast majority), the correspondance between genre and shape is surprisingly good -- this of course being the main result of the paper.

The paper reports on work in progress, but given the article format. Answers to the next question should thus be expected to come in a separate publication, perhaps in another conference series. But the present article deserves it place where it is.


**Paper Type:**

Long paper

---

### Official Review · Reviewer_UUh8 · 2023-03-09
**The paper tries to find a formula which would evaluate the lexical richness of a text, while being insensitive to the length of the text. It does not succeed, but provides questions worth discussing.**

**Rating:** 7
**Confidence:** 3

**Review:**

The paper tries to find a formula which would evaluate the lexical richness of a text, while being insensitive to the length of the text.
It does not succeed, but as a side effect shows that corpora might be characterized and meaningfully compared by some type-token ratio measure.
The paper thus contributes to the theme of understanding what exactly varies when we are dealing with different corpora.

The main strength of the paper lies in its theme.
The main weakness lies in its inability to arrive at a concrete positive result.

Specific remarks:

How does the paper define "word": is it an item of vocabulary or an item of text? Are "fisk" and "fiskar" two different words or two different forms of the same word "fisk"? Depending on the choice, the type count of a text would be different. The question may be relevant in case TTR of corpora are influenced by  different factors: one may have more variety in grammatical forms (because it contains complex grammatical constructs), the other may have a larger vocabulary (because it contains articles dealing with many different subjects).

Kalmar10 seems to have the highest lexical richness (on figures 2 to 11). Actually, this is obviously because the texts have been digitized by OCR and thus they contain many errors, i.e. there is a lot of variation in the digital orthography of tokens that represent the same word. The same is observable on figure 12 where the lexically most "rich" corpora are collections from beginning of the 20th century.

gp1994 behaves weirdly on figures 2 to 11, indicating that one should look at it and pay attention to its compilation process; this is a valuable hint by the paper.

Section 3 presents various graphs showing how some formula relating type and token counts gives different results for different corpora and corpus sizes. As expected, the position of the curves with regard to each other stays the same. So this section shows a simple mathematical regularity that the results are determined by the data, no matter how you change the way you calculate them.

Section 4 suggests improvements, aiming at finding a formula that would result in an horizontal line for each corpus, i.e. a formula that would eliminate  dependence on the corpus size. However, the author is not satisfied with any of the resulting formulas - the corrections to the formulas either seem ad hoc for namely this set of corpora, or the corrections result in non-horizontal lines. I agree with the authors in this evaluation.

Section 5 is incomprehensible for me. I cannot understand what the figures are telling us, or whether they are telling anything meaningful at all.

Summary:

The paper tries to find a formula which would evaluate the lexical richness of a text, while being insensitive to the length of the text. It does not succeed, but provides questions worth discussing.

**Paper Type:**

Long paper

---

### Official Review · Reviewer_aCrp · 2023-03-10
**Ambitious survey, but lacking in discussion and result analysis**

**Rating:** 6
**Confidence:** 4

**Review:**

The authors review existing measures of lexical richness, examine their length dependence, and propose various improvements to the measures.

The major positive takeaway is that the paper gives a thorough review of measures of lexical richness, and demonstrates their applicability to different genres and lengths of text.

However, there are some drawbacks to how the analysis is presented, and what the goals of the improvements should be. First, there is no description of the corpora used in Sections 3 and 4, without cross-referencing the curve colours with the colour-coding in Figure 12. The paper would benefit from a clearer description of the content and/or genre in Section 2.

Second, the colour-coding itself is an issue, given that the colour scheme is neither greyscale- nor colourblind-friendly. Ideally, the graphs should help differentiate between the corpora by using different markers and/or line styles. The observations related to Figure 12 are quite interesting, and making the graph more universally readable would be a great benefit.

Third, and most importantly, the motivation for the improvements is unclear. Reading through Section 3, what is the reasoning behind the insistence that the measures' curves should not overlap (or should/shouldn't adhere to some particular shape)? If this should be clear from previous work, then this should be reflected upon in the background overview. As it is, by the end of Section 3 there is a strong feeling that science is being done backwards -- tailoring measures towards producing the graphs we want to see. Perhaps the purpose of the experiments would be clearer if there was a more substantial discussion of the results, rather than just a verbal description of the figures with no further interpretation, which continues in Section 5. In Section 4, line 438, it is still unclear what are the criteria that make for a good or better result.

Section 5 covers further interesting experiments, but again, it would be helpful if the datasets used were clearly described somewhere in the text, and connected to the filenames listed in the figures.

Finally, a minor issue is the unclear writing in certain portions of the paper. At 093, when speaking of large and homogeneous corpora -- how large? Homogeneous in which sense? Same in 197 -- different how? in 207, why do we safely assume that values below 1000 may be ignored? In 316, do the further tests and other corpora refer to the experiments in Sec. 5?

To summarise, the paper presents an ambitious and thorough survey, but requires some rewriting to show the motivation behind the research and help interpret the results.

**Paper Type:**

Long paper

---

### Decision · Program_Chairs · 2023-03-17

Accept